# Female Basketball Players’ Jump and Sprint Performance After Plyometric Jump Training Compared to Resistance Training

**DOI:** 10.3390/sports13110374

**Published:** 2025-11-01

**Authors:** Yuhang Tian, Kai Xu, Wenxuan Fang, Rodrigo Ramirez-Campillo

**Affiliations:** 1School of Athletic Performance, Shanghai University of Sport, No. 200, Henren Road, Shanghai 200438, China; tyh18736360835@163.com (Y.T.); 2221152066@sus.edu.cn (K.X.); 2School of Medical and Health Sciences, Centre for Human Performance, Edith Cowan University, Joondalup 6027, Australia; 3Exercise and Rehabilitation Sciences Institute, School of Physical Therapy, Faculty of Rehabilitation Sciences, Universidad Andres Bello, Santiago 7591538, Chile; ramirezcampillo@gmail.com; 4Sport Sciences and Human Performance Laboratories, Instituto de Alta Investigación, Universidad de Tarapacá, Arica 1010069, Chile

**Keywords:** human physical conditioning, plyometric exercises, muscle strength, athletic performance, sports, team sports

## Abstract

Background: Plyometric training (PT) and resistance training (RT) can improve jumping and sprinting performance, although their comparative effectiveness in elite female basketball players remain unknown. Objectives: To compare the effects of PT and RT on jumping and sprinting performance in elite female basketball players. Methods: Thirty elite female basketball players were randomly assigned to PT (*n* = 10), RT (*n* = 10) or control groups (*n* = 10, standard basketball training). Performance assessments before and after the interventions (8 weeks, 16 training sessions) included countermovement jump (CMJ) height and peak power, drop jump (DJ) height and reactive strength index (RSI), standing long jump distance (LJ), CMJ with arm swing (CMJA) height, running CMJA height, and 22.2 m linear sprint time. Performance changes were analyzed using linear and Bayesian mixed-effects models. Results: Compared to controls, RT and PT improved the RSI. Additionally, PT improved (*p* < 0.05; posterior probability >0: 99.4–99.9%) CMJ height, CMJ peak power, DJ height and RSI, LJ, CMJA, Running CMJA and sprint time when compared to both controls and RT. Conclusions: Compared to RT, PT induced greater jumping and sprinting performance improvements in elite female basketball players.

## 1. Introduction

Basketball is a multidirectional, intermittent team sport [1,2]. The ability to generate high levels of muscular power in a short period of time is a key determinant of athletic performance [3,4]. This capacity is important in executing sport-specific movements such as jumping, linear sprinting, acceleration, deceleration, and change in direction, which are repeatedly performed by players in both offensive and defensive situations [1,5,6]. Among these, jumping and linear sprinting are among the most frequently executed movements throughout a basketball game and are critical for scoring and rebounding success [7]. Therefore, developing jumping and sprinting abilities is essential for enhancing the competitive performance of basketball players [1,2].

Plyometric training (PT) and resistance training (RT) are widely regarded as safe and effective strategies for improving jumping and sprinting performance in basketball players [8,9,10,11]. PT involves a rapid pre-stretch of the muscles (eccentric phase) that triggers the stretch reflex, immediately followed by an explosive contraction (concentric phase) [12,13]. This method relies on the storage of elastic energy during rapid passive stretching and the activation of neuromuscular responses, thereby enhancing force production during the subsequent concentric contraction and effectively improving jump and sprint performance [14,15]. In contrast, RT primarily enhances neuromuscular force-generating capacity and improves the force component of the force–velocity profile, which contributes to better performance in high-intensity movements [16,17]. The training adaptations of RT are not limited to potential muscle hypertrophy but also include greater recruitment of high-threshold, fast-twitch motor units, improved intermuscular coordination, and increased motor unit discharge rate, all of which facilitate greater force output within a short period [18,19].

Building upon the theoretical mechanisms underlying PT and RT, previous studies have examined their practical effects on athletic performance. Whitehead et al. [20] compared the effects of 8 weeks PT and RT on jumping and sprinting performance in adult males. Their results indicated that the PT group significantly improved vertical jump performance, while no significant differences were observed between the groups for linear sprint performance. In contrast, Negra et al. [19] found that after 12 weeks of training, the RT group (0.66 to 1.21) showed greater effect sizes than the PT group (0.50 to 1.03) for both vertical jump and linear sprint performance in adolescent soccer players. Other studies have also reported inconsistent or similar effects of PT and RT on jumping and sprinting performance [21,22,23,24].

Despite the growing body of evidence, most existing studies have focused predominantly on adolescent or adult male athletes, leaving a critical gap in understanding how these interventions affect elite female basketball players [21,22,23,24]. Moreover, the absence of control groups in many studies limits the ability to attribute observed effects specifically to PT or RT [22,23,25,26]. More importantly, many of the existing findings rely primarily on *p*-values, which can be misleading, especially in studies with small sample sizes [8,9].

Therefore, this study aimed to investigate the effects of PT, RT, and a control condition on jumping and sprinting performance in elite female basketball players. To enhance the robustness of the findings, both linear mixed-effects models and Bayesian approaches were applied. We hypothesized that eight weeks of PT would lead to greater improvements compared to resistance training and the control condition.

## 2. Materials and Methods

### 2.1. Participants

Using G*Power software (version 3.1.9.7, Düsseldorf, Germany), the a priori sample size estimation with repeated-measures ANOVA (2 measurements × 3 groups) indicated 30 total participants to detect a Cohen’s f effect size (ES) = 0.25, assuming a correlation among repeated measures of 0.6, a power of 80%, and α = 0.05.

Participants recruitment was conducted in a professional basketball team from a local university. Before the experiment began, voluntaries were verbally informed about the study procedures, potential risks, and benefits. Furthermore, participants were told that they should be able to maintain their normal dietary and sleep habits throughout the study period, and to refrain from using any supplements or stimulants within 24 h prior to each training and/or assessment session. All participants provided written informed consent prior to participation. This study was approved by the Ethics Committee of Shanghai University of Sport (ID number: 102772024RT070).

Female basketball players voluntaries (*n* = 33) from Shanghai University of Sport were screened according to inclusion-exclusion criteria: (i) >10 years of basketball training experience, (ii) national- or higher-level competitions within the past year; (iii) no lower limb injuries within the past six months; (iv) free from health issues. After screening, 30 athletes were included.

To minimize the impact of basketball playing position differences on the intervention effects, participants were stratified as guards (*n* = 12), forwards (*n* = 12), or centers (*n* = 6). Subsequently, using an online randomization tool (http://www.randomizer.org (accessed on 11 September 2025)), participants were assigned to PT (*n* = 10), RT (*n* = 10) and control training (*n* = 10). Table 1 provides participants’ descriptive characteristics before the interventions. All participants were categorized as elite/international (tier 4) [27]. To reduce potential measurement bias, the assessors responsible for performance testing were blinded to group allocation.

### 2.2. Training Intervention

Training was held indoors (i.e., PCV sports floor) under the same conditions (22 °C to 26 °C, no wind), at fixed times on Mondays and Thursdays (2:45–4:45 PM). All training sessions were supervised by two CSCS coaches, and preceded by a standardized 20 min warm-up, including (i) general warm-up (e.g., jogging, high knee running, butt kicks), (ii) static and dynamic stretching focusing on thigh-related muscle groups, (iii) dynamic exercises (e.g., small step running, skipping, ladder drills), (iv) short-distance sprints.

The PT and RT groups substituted a portion of their existing conditioning sessions with the intervention protocols (Table 2), while maintaining the rest of their regular training schedule. Each intervention consisted of 2 sessions per week for 8 weeks. The control group continued their regular training. The PT and RT exercises target vertical, horizontal, bilateral, and unilateral force production patterns. Training volume and intensity progressively increased over the intervention period, with a scheduled reduction in training load during week 8. This design was intended to replicate the functional requirements of basketball players, particularly the ability to generate explosive force for jumping and sprinting. In both groups, training load progression was achieved primarily through increased sets/repetitions and/or external resistance. Total training volume for PT was 1300 jump contacts and 435 m of sprinting, comprising 140–176 jumps plus 45–60 m sprint distance per session. For RT, total training volume included 888 repetitions of RT exercises and 435 m of sled pushing, with 96–120 reps + 45–60 m sled pushing distance per session (Table 2). Recovery time between sets was between 15 s and 60 s, and 3 min to 4 min between exercises.

Due to the menstrual cycle-related issues, two participants missed one session, four missed two sessions, and five missed three sessions. However, all participants maintained overall attendance > 80%.

### 2.3. Physical Performance Tests

Two weeks prior to testing participants underwent two familiarization sessions to minimize learning effects. One week before and after interventions the participants were assessed for jump and sprint performance. Jump tests were performed on a jump mat system (Smart Jump, Fusion Sport, Australia) [28], and linear sprint time was measured using two pairs of photoelectric gates (Smart Speed, VALD Performance) placed at the start and finish lines [29]. All jump and sprint tests were conducted on the same day. If participants experienced menstrual cycle-related symptoms that hinder their maximal effort-motivation during tests, these were rescheduled. Each jump and sprint test were performed three times, with 1 min of rest between jump trials and 2 min between sprint trials. The average value across the three attempts was used for analysis [30]. A 4 min rest was provided between different tests. Verbal encouragement was provided for all tests to ensure maximal effort.

#### 2.3.1. Countermovement Jump (CMJ)

Participants stood upright on the jump mat with their hands on their hips. From this position, they performed a rapid downward movement (~110° knee flexion) followed by an immediate vertical jump. During take-off, the knees remained extended (i.e., no knee tuck was allowed). The landing had to occur fully within the mat area. The jump height and peak power were recorded.

#### 2.3.2. Drop Jump (DJ)

Participants stood on a 30 cm box placed adjacent to the jump mat. With their toes close to the edge of the box, they stepped off and landed with both feet on the mat, immediately performing a maximal vertical jump with minimal ground contact time. The resulting measures included jump height and RSI [jump height (m) ÷ ground contact time (s)].

#### 2.3.3. Standing Long Jump (LJ)

Participants stood behind a marked line and performed a maximal horizontal jump. The horizontal distance from the take-off line to the landing point (i.e., heels) was recorded as the valid score. Any faults such as stepping over the line, asynchronous landing, falling backward, or using hands for support, invalidated the attempt.

#### 2.3.4. CMJ with Swing Arm (CMJA)

Participants stood upright with feet shoulder-width apart and performed a countermovement jump using a coordinated arm swing and hip/knee flexion. During the jump, one arm was fully extended to reach the highest possible point on a vertical measuring device (CSSIT, Vertical jump trainer, China). A soft landing with knee flexion was required.

#### 2.3.5. Running CMJA

Participants started their approach from a self-selected distance within the basketball court, beginning inside the three-point line (6.75 m from the basket). They performed a running approach and jumped off one foot (either left or right), using one arm to reach the highest possible point on a vertical jump measurement device (CSSIT, Vertical jump trainer, China).

#### 2.3.6. Linear Sprint

The test started at the baseline of one side of a basketball court and ended at the free-throw line on the opposite side (22.2 m), distance equivalent to three-quarters of a basketball court.

### 2.4. Statistical Analysis

Statistical analysis was performed by the author K.X., who was not involved in experimental procedures, i.e., blinded to group allocation. Data are presented as mean ± standard deviation (SD). The reliability and consistency of CMJ height were assessed using the coefficient of variation (CV) and the intraclass correlation coefficient (ICC) [31], calculated as ICC (3,1) (two-way mixed-effects model, consistency type).

To examine performance changes from baseline within each group (i.e., PT, RT, and control), as well as between-group differences and interaction effects, linear mixed-effects models were employed using the *lmerTest* package (*R* version 4.3.0; R Core Team, Vienna, Austria), with subjects treated as a random effect [32]. *p*-values were derived from Type III ANOVA with Satterthwaite’s method, and post hoc pairwise comparisons were performed using the *emmeans* package [33]. To reduce reliance on the binary interpretation of *p*-values, Bayesian mixed-effects models were employed to estimate the probability of performance improvements from baseline within each group. Posterior distributions of performance changes, along with their 95% credible intervals (CrI), were derived using the brms package [34], assuming Gaussian likelihoods and default weakly informative priors. This Bayesian framework offers advantages in small-sample settings by providing richer probabilistic inferences beyond traditional null hypothesis significance testing.

Additionally, effect sizes (ESs) and post hoc statistical power were calculated to further interpret the results [35,36]. Hedges’ g (adjusted for small sample bias) was used to determine the magnitude of observed effects, with thresholds of <0.2, 0.2–0.5, 0.5–0.8, and >0.8 representing trivial, small, moderate, and large effects, respectively. Statistical significance was set at *p* < 0.05.

## 3. Results

The reliability of the performance tests is presented in Table 3.

### 3.1. Jump Performance

After 8 weeks of training, the PT group showed significant improvements in CMJ height (ES = 1.18, power = 91.35%, Figure 1a), CMJ peak power (ES = 1.39, power = 97.32%, Figure 1b), DJ (ES = 0.91, power = 73.00%, Figure 1c), RSI (ES = 1.77, power = 99.85%, Figure 1d), LJ (ES = 1.06, power = 84.40%, Figure 1e), CMJA (ES = 1.32, power = 95.80%, Figure 1f), and Running CMJA (ES = 1.22, power = 92.77%, Figure 1g). In contrast, the RT group showed a significant improvement only in RSI (ES = 0.91, power = 72.81%), while no significant changes were observed in the control group for any of the jump tests (Table 4).

Linear mixed-effects models revealed significant group × time interactions for CMJ height (*p* = 0.01), CMJ peak power (*p* = 0.01), DJ (*p* = 0.049), RSI (*p* < 0.01), and CMJA (*p* = 0.04). Significant main effects of group were also found for all performance outcomes (*p* < 0.05). Post hoc comparisons showed that the PT group outperformed the control group in all jump tests (*p* < 0.05). Except for Running CMJA, the PT group also outperformed the RT group in all other jump tests (*p* < 0.05). The RT group was superior to the control group only in CMJA (*p* = 0.02).

All participants (100%) after PT improved jump performance (Figure 2), the RT group demonstrated improvement in all outcomes for 100% of participants (except RSI, 90%). In contrast, improvement rates in the control group ranged from 20% to 70%.

### 3.2. Sprint Performance

After 8 weeks of training, only the PT group demonstrated a significant improvement in three-quarter court sprint performance (ES = −2.20, power = 99.99%, Figure 1h). No significant changes were found in either the RT or control groups (*p* = 0.76, Table 4). Linear mixed-effects models indicated a significant main effect of group (*p* = 0.04) and a significant group × time interaction (*p* < 0.01). Post hoc analysis revealed that the PT group showed significantly greater improvements than both the RT and control groups (*p* < 0.01).

Based on individual data (Figure 2), both the PT and RT groups improved sprint performance in all participants (100%), whereas the control group showed improvements in 40% of participants.

## 4. Discussion

To the best of our knowledge, this study is the first to investigate the effects of PT, RT, and a control on jumping and sprinting performance in elite female basketball players. Consistent with our hypothesis, the PT group showed significant improvements in vertical jump, horizontal jump, and linear sprint performance, while no significant changes were observed in the RT and control groups. Furthermore, PT was more effective than RT and control in enhancing CMJ height, CMJ peak power, DJ, RSI, LJ, CMJA, Running CMJA, and three-quarter court sprint performance. These improvements, reflected by large effect sizes, translate into practical gains. For example, increases in CMJ height or sprint performance can allow these athletes to move faster during offense and defense, potentially improving overall scoring efficiency on the court. These findings strongly support the effectiveness of PT in improving jumping and sprinting performance in elite female athletes.

This study considered progressive load increases when designing the PT and RT groups, with a reduction in training load implemented during the final week. Additionally, the training programs for both PT and RT groups included unilateral and bilateral vertical training, unilateral and bilateral horizontal training, and lateral training. The current results indicate that the PT group showed significant advantages across various jump tests and the linear sprint test. A meta-analysis by Stojanović et al. [37] demonstrated that PT is more effective in improving jump performance utilizing the stretch-shortening cycle (e.g., CMJ and DJ), as PT enhances the elastic properties of the muscle-tendon unit and the neural activation and firing rate of related motor units [1,14,15]. Similarly, all the jump tests in this study involved stretch-shortening cycles, which may explain the superior performance observed in the PT group [1,8,9]. However, it should be noted that both the PT and RT programs included a relatively low number of sets for multi-joint exercises, which may have limited the overall effectiveness of the RT program. Therefore, the apparent superiority of PT over RT should be interpreted with caution.

In contrast, the RT group focused more on improving the force component of the force-velocity curve, especially at lower speeds, which does not align well with the rapid force production required for high-speed movements [4,10,11]. This mismatch may be one reason why RT led to a positive but non-significant improvement in jump performance in this study. Additionally, although both PT and RT groups included sprint training, only the PT group showed a significant improvement in sprint performance, and its effects were notably superior to those of RT and control. Given that the elite female basketball players in this study had high levels of strength (with an average back squat close to double their body weight), further resistance training might yield marginal gains [38]. However, utilizing more specific training that closely mirrors the movements involved in jumping and sprinting is more effective in converting existing strength into dynamic power. This also explains why the PT group showed improvements of 13.1%, 10.6%, 4.8%, 3.8%, 4.1%, and 6.5% in CMJ, DJ, LJ, CMJA, Running CMJA, and three-quarter court sprint tests, respectively, which can translate into practical advantages on the court. For example, greater jump height can provide an advantage in rebounding, faster acceleration and enhanced jumping ability can facilitate scoring on offense, and quicker movement speed can increase defensive pressure on opponents.

Female basketball players that only engage in regular training showed performance decline. Although this decline did not reach statistical significance, the results suggest that a single training modality is insufficient to maintain the performance of elite athletes [39]. This finding underscores the importance of adopting a variety of training formats and loads for elite female basketball players, avoiding reliance on a single training approach. Additionally, when comparing the PT and RT groups to the control group, we found that the PT group showed significantly greater improvements in jump and sprint performance compared to the control group, while the RT group only showed superior performance in RSI compared to the control group. Despite the small sample size (only 10 participants per group), the PT group demonstrated high statistical power (all >80%) and improved the performance of all athletes on an individual level.

Notably, PT produced large effect sizes for jump and sprint performance (ES = 0.91 to 2.20), indicating substantial practical improvements. These pronounced effects are likely due to the high specificity of PT, which closely mimics the explosive demands of basketball. PT enhances the stretch–shortening cycle, neuromuscular efficiency, and the utilization of elastic energy, thereby facilitating a more effective transfer of existing maximal strength into dynamic performance [40,41]. In contrast, although RT can increase maximal force production, its transferability to high-velocity movements depends heavily on training specificity [3,11]. The limited improvements in sprint and jump performance observed in the RT group may therefore be attributed to a lack of task-specific neuromuscular adaptations [3,11]. As a result, the PT group outperformed the RT group across all jump and sprint tests, further highlighting the advantages of PT in enhancing the athletic performance of elite female basketball players.

### 4.1. Potential Limitations

The PT and RT protocols were designed to be as comparable as possible, including exercises motor patterns (e.g., CMJ and sprints for PT vs. squats and sled pushes for RT) and total training duration, the inherent differences between PT and RT made it difficult to precisely match their intensity and training volume. For example, the RT group required more time compared to the PT group to perform a given repetition, thus the total number of sets was lower in the RT group. Further, the 30 participants in this study had experience in RT, although they had limited systematic experience with specific PT. Considering these factors, the training volume and intensity of the RT group in this study may have been insufficient to elicit significant improvements in jumping and sprint performance in elite female basketball players, and future studies should consider increasing training volume and intensity. Furthermore, the specificity between performance tests and PT exercises may have favored the magnitude of performance adaptations after PT compared to RT, and this was further limited by logistical constraints that preclude the research team to perform 1RM testing after the interventions.

Due to participants’ training schedules and other practical constraints, mechanistic measures of jumping and sprinting were not collected, which limits the ability to explain the underlying mechanisms of PT effects. Although the sample size in this study was small, with only 10 participants per group, it is important to note that overall sample sizes are typically limited when studying elite athletes. Moreover, post hoc statistical power analyses indicated that the results provide robust support for the effectiveness of PT (73.00–99.99%). Therefore, the small sample size does not constitute a major limitation of this study.

The moderate ICC values (<0.80) observed for the 22.2 m linear sprint may be related to the inter-trial recovery interval of 2 min used in our protocol, which was chosen based on previous sprint-testing procedures to balance adequate recovery with testing time constraints [42]. As no systematic performance decline was observed across the three trials, we used the mean value to represent the athletes’ typical performance and to minimize the influence of single-trial variability [30]. For the LJ, CMJA, and running CMJA, the ICCs were also below 0.80, likely due to the higher sensitivity of these tasks to technical execution (e.g., arm swing) and the finite resolution of the measurement devices [30]. While ICCs < 0.80 indicate moderate reliability, these values do not materially alter the interpretation of our findings, as the overall patterns were consistent across multiple measures.

### 4.2. Practical Application

For female basketball players with high levels of strength, we recommend that the PT program consists of 8 weeks of training, with two sessions per week totaling 1300 jumps and 435 m of sprinting. Each session includes 140–176 jumps and 45–60 m of sprinting. Progressive load increases with a final taper are unlikely to increase injury risk in this population. Recovery intervals are recommended to be 15–60 s between sets and 3–4 min between exercises. In contrast, for RT group, a higher training volume and/or intensity than that applied in this study may be required to achieve meaningful performance gains.

## 5. Conclusions

A progressive PT program, combining bilateral, unilateral, and sprint exercises, effectively improved vertical and horizontal jump performance, as well as linear sprinting performance, in elite female basketball players after 8 weeks and 16 training sessions. In comparison, RT showed a positive effect on jumping and sprinting performance but did not reach statistical significance, with the probability of a performance improvement greater than 0 ranging from 73.5% to 90.3%. Additionally, the PT group demonstrated significantly greater improvements in vertical and horizontal jump performance, as well as linear sprinting, compared to both the RT and control group. Therefore, coaches and athletes can adopt the PT design methods used in this study to effectively enhance the jumping and sprinting performance of elite female basketball players.

## Figures and Tables

**Figure 1 sports-13-00374-f001:**
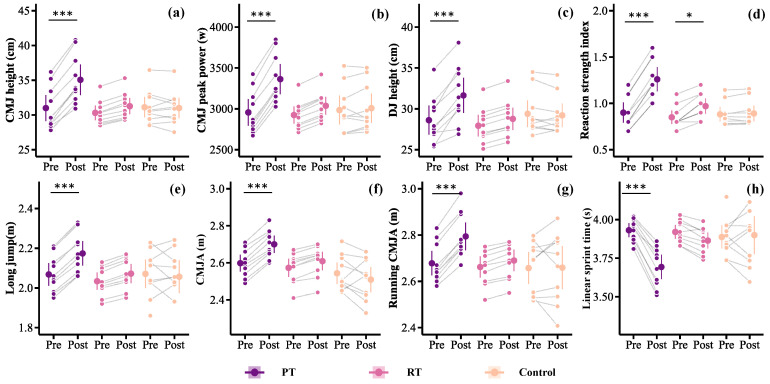
Changes in performance between pre- and post-tests for the three groups after eight training sessions. (**a**) CMJ height; (**b**) CMJ peak power; (**c**) DJ height; (**d**) Reactive strength index; (**e**) long jump; (**f**) CMJA: (**g**) Running CMJA; (**h**) 22.2 m linear sprint time; CMJ, countermovement jump; DJ, drop jump; CMJA, CMJ with swing arm; PT, plyometric training; RT, resistance training; *** *p* < 0.01; * *p* < 0.05.

**Figure 2 sports-13-00374-f002:**
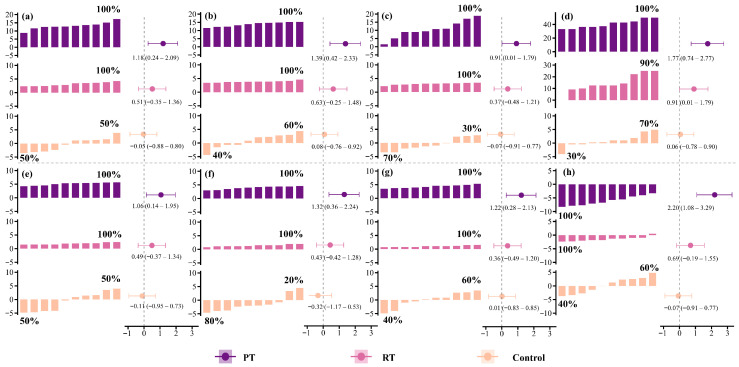
Individual changes and effect sizes before and after 8 weeks of training in three groups. (**a**) CMJ height; (**b**) CMJ peak power; (**c**) DJ height; (**d**) Reactive strength index; (**e**) long jump; (**f**) CMJA: (**g**) Running CMJA; (**h**) 22.2 m linear sprint time; CMJ, countermovement jump; DJ, drop jump; CMJA, CMJ with swing arm; PT, plyometric training; RT, resistance training.

**Table 1 sports-13-00374-t001:** Participants’ descriptive characteristics.

Variable	PT (***n*** = 10)	RT (***n*** = 10)	Control (***n*** = 10)
Age (y)	19.6 ± 0.3	19.6 ± 0.4	19.1 ± 0.7
Training experience (y)	11.3 ± 0.3	11.9 ± 0.4	12.3 ± 0.95
Height (m)	174.2 ± 2.4	174.6 ± 2.1	173.3 ± 6.4
Weight (kg)	64.2 ± 1.7	64.9 ± 1.3	64.3 ± 4.5
Squat 1RM (kg) *	116.0 ± 3.6	116 ± 3.6	117.5 ± 12.1

PT = Plyometric training; RT = resistance training; 1RM: one repetition maximum. This measure was performed only at baseline, and the RT group used this measure to control the loads for main training exercises; * Represents the participants’ one-repetition maximum (1RM) in the squat.

**Table 2 sports-13-00374-t002:** Training content for the PT and RT groups.

Group	Exercise and Load *	Sets × Repetitions or Distance		Rest **
		Weeks 1, 3, 5	Weeks 2, 4, 6, 7, 8	
PT	CMJ	3 × 8	3 × 10	15 s
	Left leg CMJ	2 × 6	2 × 8	
	Right leg CMJ	2 × 6	2 × 8	
	Lateral skater jumps	3 × 10	3 × 12	
	30 cm hurdle jump	3 × 8	3 × 10	
	30 cm drop jump	3 × 6	3 × 8	15–30 s
	30 cm left leg drop jump	2 × 5	2 × 6	
	30 cm right leg drop jump	2 × 5	2 × 6	
	Linear sprint	3 × 15 m	3 × 20 m	30 s
RT	Back squat Week 1–2: 60% 1RM; Week 3–4: 70% 1RM; Week 5–7: 75% 1RM; Week 8: 60% 1RM	2 × 8	2 × 10	3–4 min
	Left and right leg Bulgarian squat Week 1–2: 40% 1RM; Week 3–4: 50% 1RM; Week 5–7: 55% 1RM; Week 8: 40% 1RM	1 × 6	1 × 8	
	Seated hip abduction Week 1–2: 10 kg; Week 3–4: 15 kg; Week 5–7: 20 kg; Week 8: 10 kg	3 × 10	3 × 12	30 s
	Keiser air squat Heel raise Week 1–2: 50% 1RM; Week 3–4: 60% 1RM; Week 5–7: 70% 1RM; Week 8: 50% 1RM	2 × 8 2 × 6	2 × 10 2 × 8	3–4 min
	Left and right leg heel raise Week 1–2: 30% 1RM; Week 3–4: 40% 1RM; Week 5–7: 50% 1RM; Week 8: 30% 1RM	1 × 5	1 × 6	
	Sled-push Week 1–2: 40% 1BM; Week 3–4: 60% 1BM; Week 5–7: 80% 1BM; Week 8: 40% 1BM	3 × 15 m	3 × 20 m	1 min

**, The recovery time between each pair of training exercises is 3–4 min; Keiser air squat: lean torso ~30°, feet positioned forward to simulate sprint-start posture. PT, plyometric training; RT, resistance training; CMJ, countermovement jump; BM, body mass. *, for PT exercises no external load was used at weeks 1, 2 and 8, while in weeks 3 and 4 varied from 2 to 4 kg, weeks 5 and 6 from 4 to 8 kg, and at week 7 from 6 to 10 kg.

**Table 3 sports-13-00374-t003:** Dependent variable’s reliability.

Variable	ICC [95%CI]	CV
CMJ height	0.93 [0.89–0.95]	2.45%
CMJ peak power	0.94 [0.91–0.96]	2.55%
DJ height	0.95 [0.92–0.97]	4.20%
Reactive strength index	0.99 [0.98–0.99]	6.40%
Standing long jump	0.79 [0.70–0.86]	5.32%
CMJ with swing arm	0.69 [0.58–0.79]	3.49%
Running CMJ with swing arm	0.73 [0.62–0.82]	5.67%
Linear sprint	0.76 [0.66–0.84]	2.25%

ICC, intraclass correlation coefficient; CV, coefficient of variation; CI, confidence interval; CMJ, countermovement jump.

**Table 4 sports-13-00374-t004:** The effects of three intervention methods on different athletic performances.

Group	Pre	Post	Change [95%Crl]	*p*	ES	Power	Probability
Countermovement jump height (cm)							
PT	30.99 ± 3.01	35.06 ± 3.55	4.06 [2.10–5.99]	**0.00**	1.18	91.35%	99.99%
RT	30.30 ± 1.71	31.26 ± 1.89	0.96 [−0.98–2.94]	0.32	0.51	30.33%	83.30%
CT	31.11 ± 2.38	31.00 ± 2.37	−0.11 [−2.04–1.77]	0.91	−0.05	5.19%	45.90%
Countermovement jump peak power (w)							
PT	2956.10 ± 263.54	3362.90 ± 296.40	406.95 [233.46–578.18]	**0.00**	1.39	97.32%	99.99%
RT	2923.50 ± 169.63	3037.00 ± 177.07	113.44 [−57.52–287.88]	0.19	0.63	42.52%	90.30%
CT	2984.35 ± 285.33	3007.57 ± 289.41	21.46 [−153.56–194.90]	0.79	0.08	5.55%	60.20%
Drop jump (cm)							
PT	28.61 ± 2.86	31.64 ± 3.46	3.04 [1.22–4.91]	**0.00**	0.91	73.00%	99.99%
RT	27.93 ± 2.13	28.77 ± 2.20	0.85 [−1.03–2.77]	0.36	0.37	18.33%	81.50%
CT	29.39 ± 2.63	29.20 ± 2.33	−0.19 [−2.09–1.68]	0.84	−0.07	5.47%	42.30%
Reactive strength index (m/s)							
PT	0.90 ± 0.18	1.26 ± 0.21	0.36 [0.24–0.47]	**0.00**	1.77	99.85%	99.99%
RT	0.85 ± 0.12	0.97 ± 0.13	0.12 [0.01–0.24]	**0.04**	0.91	72.81%	98.00%
CT	0.88 ± 0.13	0.89 ± 0.14	0.01 [−0.11–0.12]	0.87	0.06	5.38%	56.60%
Long jump (m)							
PT	2.07 ± 0.09	2.17 ± 0.10	0.11 [0.03–0.18]	**0.01**	1.06	84.40%	99.60%
RT	2.03 ± 0.07	2.07 ± 0.08	0.04 [−0.04–0.12]	0.32	0.49	28.47%	83.40%
CT	2.07 ± 0.12	2.06 ± 0.13	−0.02 [−0.09–0.06]	0.71	−0.11	6.12%	34.80%
Countermovement jump with swing arm (m)							
PT	2.60 ± 0.07	2.70 ± 0.07	0.10 [0.03–0.18]	**0.01**	1.32	95.80%	99.70%
RT	2.57 ± 0.08	2.61 ± 0.08	0.04 [−0.04–0.11]	0.32	0.43	23.32%	83.30%
CT	2.54 ± 0.09	2.51 ± 0.11	−0.03 [−0.11–0.04]	0.34	−0.32	15.09%	18.50%
Running countermovement jump with swing arm (m)							
PT	2.68 ± 0.09	2.79 ± 0.10	0.12 [0.03–0.21]	**0.01**	1.22	92.77%	99.40%
RT	2.66 ± 0.07	2.69 ± 0.07	0.03 [−0.06–0.12]	0.53	0.36	17.50%	73.50%
CT	2.66 ± 0.11	2.66 ± 0.15	0.00 [−0.09–0.09]	0.96	0.01	5.02%	51.80%
22.2 m linear sprint time (s)							
PT	3.93 ± 0.07	3.69 ± 0.13	−0.24 [−0.34–−0.14]	**0.00**	−2.20	99.99%	99.99%
RT	3.92 ± 0.07	3.86 ± 0.08	−0.06 [−0.16–0.04]	0.27	−0.69	49.56%	86.40%
CT	3.89 ± 0.12	3.90 ± 0.20	0.01 [−0.09–0.11]	0.81	0.07	5.44%	40.80%

PT, plyometric training; RT, resistance training; CT, control training; Crl, confidence interval; *p*-values were obtained based on post hoc comparisons using linear mixed effects model; power indicates the statistical power of the posterior; the probability is derived from a Bayesian mixture model, showing the probability of a result >0, where sprint is the probability of <0.

## Data Availability

The data used to support the findings of this study are available from the corresponding author upon request.

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
