# Peer review of "Female Basketball Players’ Jump and Sprint Performance After Plyometric Jump Training Compared to Resistance Training"

_sports, 2025, doi:10.3390/sports13110374_

Round 1

Reviewer 1 Report

Comments and Suggestions for Authors

General commetns.

This is an interesting and timely study that addresses a clear gap: the comparative effects of plyometric (PT) and resistance training (RT) in elite female basketball players. The introduction is well written and situates the work within the existing literature, making a persuasive case for the need to study this population. The experimental design is relatively strong, with random group allocation and a control group, which helps strengthen causal inference. However, several aspects limit the persuasiveness of the findings. Most notably, while statistical significance is carefully reported, effect sizes and mechanisms of change are underdeveloped. The discussion also misses the opportunity to contextualize findings with practical applications for coaches.

Specific Comments

Methods

The athletes were randomly assigned to PT, RT, and control groups, which is a strength. However, participants were aware of the training content, meaning expectancy effects could have influenced results. A more detailed explanation of blinding (e.g., of assessors) would help reassure readers.

Pre–post assessments were appropriate, but the study does not sufficiently explore mechanisms of change. For example, measures of neuromuscular adaptation, tendon stiffness, or rate of force development could have provided insight into why PT outperformed RT. Both PT and RT interventions are described in detail. Still, more justification is needed for why specific exercises were chosen and how they reflect optimal programming for basketball players. At present, the rationale for the selection feels more descriptive than persuasive.

The statistical analysis is comprehensive (mixed models and Bayesian approaches), but the emphasis is still on p-values. Although Hedges’ g effect sizes are reported, they are not adequately interpreted in the text. For example, the reported ES for CMJ peak power (1.39) is large, but the source of this magnitude is not discussed.

Results

The tables are detailed, but Figure 1 is hard to interpret. The labeling is dense, and the visual presentation does not clearly show the differences between groups. A simpler graphic emphasizing between-group contrasts would be more effective.

The results section is mostly descriptive. Stronger integration of the effect sizes and probabilities into the narrative would improve interpretation. At present, readers are left with many numbers but limited synthesis. This makes it hard work for the reader. It should be easy! 

Discussion

The discussion correctly highlights that PT was more effective than RT in this population. However, the argument leans heavily on descriptive findings rather than explanatory mechanisms. Greater engagement with theories of transfer of training (force–velocity relationship, task-specific neuromuscular adaptation) would strengthen the paper. Why the intervention works should be a question running through the study. 

The practical implications are underdeveloped. For example, what are the time or load demands of PT vs RT, and how might these integrate into weekly training schedules for elite female players? Are there injury risks or compliance considerations?

Limitations are acknowledged (e.g., comparability of PT vs RT volumes, reliability of some tests), but the small sample size and limited generalizability could be stressed further. 

Recommendations for Improvement

Effect sizes – Provide clear interpretation of the Hedges’ g values. Rather than just reporting, explain whether these represent small, moderate, or large improvements and what this means practically (e.g., a 4 cm CMJ gain).

Mechanisms – Discuss in more detail why PT produced larger gains than RT. Could tendon stiffness, stretch–shortening cycle efficiency, or existing strength levels of participants explain this pattern?

Figures – look the figures and revise for interpretation. 

Practical Application – Add a subsection on how coaches might implement PT in elite female basketball training, considering volume, intensity, recovery, and safety.

Justification of Programs – Strengthen the rationale for exercise selection in both PT and RT conditions.

Transparency of Effect Sizes for Power – The paper frequently mentions “power = 90%+,” but it is not clear whether this refers to post-hoc power calculations. Clarification is needed.

In summary, I feel the paper makes a valuable contribution by providing controlled evidence on PT vs RT in elite female basketball players. The main strengths are the randomized design, inclusion of a control group, and detailed reporting of interventions. However, the persuasiveness of the conclusions is reduced by the lack of mechanistic measures, the limited interpretation of effect sizes, and insufficient discussion of practical application. Addressing these issues would significantly enhance both the scientific and applied impact of the study.

Comments on the Quality of English Language

The quality of English is generally adequate, but the manuscript would benefit from careful editing to improve clarity and flow. In several places, sentences are overly long or technical, which makes the arguments harder to follow. Simplifying phrasing, improving transitions between sections, and ensuring consistent use of terminology would enhance readability and help the research findings stand out more clearly.

Reviewer 2 Report

Comments and Suggestions for Authors

Thank you for inviting me to review this study

Line 36 "has become" why? Was it not important before?

Line 39-43 if available mention some stats regarding the number of sprints, jumps and so on estimated during a basketball game

Line 52 "force–velocity profile" it is unclear as it stands here

Line 55 "neural conduction efficiency" maybe the authors mean "discharge rate"?

Line 54 "increased motor unit recruitment" which motor units?

Line 58 "performance. [20] compared" syntax error/referencing style

Line 62 Improper referencing style

Line 62-63 report effect sizes

Line 110 "replaced a portion" unclear. Please specify the context

Table 2 the number of sets in the RT group seem relatively low given these are > national level. Needs to be consider in the discussion and limitations

"2.3. Physical performance Tests" has been repeated twice

In all physical tests verbal instructions are missing. Please provide details

Table 4 "Reaction strength index (m/s)" reactive

"Runing" running

DISCUSSION

Given the low number of sets in multijoint exercises, the authors should use caution in regards to the ineffectiveness of RT in improving ballistic performance. This needs to be discussed to have a balanced view of PT and RT

"A meta-analysis by[37]" poor referencing style

"Given that the elite female basketball players in this study had high levels of strength"... so the designed program was not adequate for this population. This needs to be discussed

Line 309-310 "the 30 participants in this study had experience in RT, although they had limited systematic experience with specific PT" so they may have benefited by a new stimulus (PT) and RT was not sufficiently dosed

Round 2

Reviewer 1 Report

Comments and Suggestions for Authors

The authors have done good job in revising the study and responding to the comments. 

Reviewer 2 Report

Comments and Suggestions for Authors

Good amendments